# Functional Verification of the Four Splice Variants from *Ajania purpurea NST1* in Transgenic Tobacco

**Hai Wang †, Xueying Hao †, Wenxin Zhang, Yuning Guo, Xiang Zhao, Yanxi Li, Wenting He, Shiyi Cai and Xuebin Song ***

College of Landscape Architecture and Forestry, Qingdao Agricultural University, Qingdao 266109, China; wh672831@126.com (H.W.); 20212210002@stu.qau.edu.cn (X.H.); 17806260856@163.com (W.Z.); 20212110007@stu.qau.edu.cn (Y.G.); crazyz18@163.com (X.Z.); lyx33091@126.com (Y.L.); qauylhwt@163.com (W.H.); qaucaishiyi@163.com (S.C.)
* Correspondence: xuebinsong@sina.cn
† These authors contributed equally to this work.

**Abstract:** *Ajania purpurea* is a small semi-shrub in the Asteraceae family. Its corolla is purplish red from the middle to the top, and its leaves and flowers are all fragrant. It can be introduced and cultivated as ornamental plants. In order to survive adversity, plants actively regulate the expression of stress response genes and transcripts. Alternative splicing is a common phenomenon and an important regulation mode of eukaryotic gene transcription, which plays an important role in various biological processes. In this study, four splice variants of the *NST1* gene were identified from *A. purpurea*, and the molecular mechanism of *NST1* alternative splice variants involved in abiotic stress was explored through bioinformatics, transgenics and paraffin sectionalization. The analysis of amino acid sequences showed that *ApNST1.1* had alternative 5′splicing, *ApNST1.2* had alternative 3′splicing and *ApNST1* had the two splicing types. The main conclusions from studying transgenic tobacco seedlings and adult seedlings under abiotic stress were as follows: *ApNST1*, *ApNST1.1* and *ApNST1.3* showed salt tolerance at seedling stage, especially *ApNST1.3*. At the mature seedling stage, the stem height of *ApNST1.1* increased significantly, and *ApNST1.1* showed obvious salt tolerance, while *ApNST1.2* showed obvious cold resistance. Compared to *Super35S::GFP*, the xylem of *ApNST1* thickened by 94 μm, and the cell wall thickened by 0.215 μm. These results are of great significance to the breeding and application of *ApNST1* to select splice variants with more resistance to abiotic stress, and to future study in this area. At the same time, they provide a new direction for *A. purpurea* breeding, and increase the possibility of garden applications.

**Keywords:** abiotic stress; alternative splicing; *A. purpurea*; *NST1*

## 1. Introduction

*Ajania purpurea* C. Shih, a small semi-shrub of the genus Asteraceae, grows in alpine gravel piles, alpine meadows and shrublands at an altitude of 4800–5300 m, and is endemic to the Gundes Mountains of Tibet. Its corolla is purple-red from the middle upward; the leaves and flowers are rich in fragrance. It displays cold and drought tolerance, and it can be introduced and cultivated as a good ornamental garden plant [1]. NAC SECONDARY WALL THICKENING PROMOTING FACTOR 1 (*NST1*) is an important NAC transcription factor for secondary wall biosynthesis, and *NST1* regulates the thickening of the secondary wall of anther endothelial cells necessary for anther cracking [2]. In addition, NAC transcription factors can affect the formation of the secondary wall through the desiccation pathway. Liu et al. found that abscisic acid regulates the formation of secondary cell walls and lignin deposition of *Arabidopsis thaliana* (L.) Heynh by phosphorylating *NST1* [3].

Adverse conditions such as cold, high temperature, salinity and drought are great threats to crop production at present, seriously affecting the growth and development, yield and quality of plants. Plants have evolved sensitive coping systems to cope with various

abiotic stresses [4,5]. NAC (NAM, ATAF and CUC) family transcription factors are a large class of plant-specific transcription factors. The N-terminus is highly conformed and is a DNA-binding region, while the C-terminus is a transcriptional activation region with great structural changes [6,7]. NAC family transcription factors are involved in the regulation of many biological processes in plants, including growth and development, organ formation and pathogen defense [8]. In addition, increasing numbers of studies have shown that NAC family transcription factors play an important role in abiotic stress responses such as drought [9], salt [10] and low temperature [11–14]. Most studies on the NAC family have focused on the resistance of crops, such as wheat, rice [8,15–17], and tomato [18,19], but the mechanism of the NAC family in *A. purpurea* remains unclear. This study will provide a preliminary analysis of *NST1*.

Alternative splicing (AS) is a common phenomenon and an important regulatory mode of eukaryotic gene transcription, which plays an important role in various physiological processes of organisms [20]. In nature, most precursor mRNAs (pre-mRNAs) transcribed from eukaryotic DNA contain exons and introns distributed in the noncoding region between exons [21,22]. Proper editing of precursor mRNAs is essential for eukaryotes. However, for many genes, the splicing of precursor mRNA is not unique. The same precursor mRNA may form multiple mature transcripts through different splicing methods, which in turn encode proteins with different functions, a phenomenon called alternative splicing. Variable splicing is a common phenomenon for most eukaryotes. Variable splicing has been reported in 61% of Arabidopsis genes, while AS events occurred more frequently under an abiotic stress [23]. There are many types of alternative splicing; the most common are the following 5 types: retained intron (RI), skipped exon (SE), alternative 5′splice site (A5SS), alternative 3′splice site (A3SS), and mutually exclusive exons (MXE). There are several other types that are less common, such as alternative promoters and alternative poly(A) [24]. Variable splicing has important biological functions. Overall, it increases the diversity of proteins encoded in organisms and regulates gene expression at the post-transcriptional level. In plants, alternative splicing has been found to play a role in many important physiological processes, including photosynthesis [25], stress response [26], flowering [27,28], and photoperiod regulation [29]. Confirming the importance of splicing in plant stress adaptation, key players of stress signaling have been shown to encode alternative transcripts, whereas mutants lacking splicing factors or associated components show a modified sensitivity and defective responses to abiotic stress.

When plants in natural conditions face harsh changes in the environment and climate, they must withstand abiotic stresses such as drought, salinity, and extreme temperatures. In order to survive adversity, plants actively regulate the expression of stress response genes and transcripts. Alternative splicing is a regulatory process by which different isomers are produced, enhancing the diversity of plant proteomes. Several studies have confirmed that alternative splicing plays an important role in plant performance, adaptability and survival. We have found the performance of the four splice variants in the purple flower under different stresses is more evidence of this view.

In our previous work, we cloned *NST1*, *SND1* and other NAC family genes, and carried out phylogenetic analysis, focusing on the evolutionary history of this wild chrysanthemum to fully understand its differentiation time. The purpose of this study was to further investigate the functional roles of four kinds of variable splice variants in the cloning of *NST1* gene, to aid in the development of resistant plants. Specifically, we studied the variable-splicing types of four splice variants, *ApNST1*, *ApNST1.1*, *ApNST1.2* and *ApNST1.3*, and observed their growth phenotypes in natural and sterile environments, including the xylic thickness of the transverse section of four splice variants in tobacco (*Nicotiana tabacum* cv. Nc89). Our ultimate aim is to provide a variety of possibilities for the selection and cultivation of *A. purpurea*, which is of great significance for the multi-faceted applications of *A. purpurea*. Furthermore, enriching the resistance breeding of *A. purpurea* is of great significance for chrysanthemum breeding.

## 2. Materials and Methods

### 2.1. Ajania purpurea Growth and RNA Isolation

The tissue culture seedlings of *A. purpurea* used in this study were derived from Research group of Professor Zhao, Beijing Forestry University, and the seedlings were grown in a growth chamber under a 16 h light and 8 h dark photoperiod at 25 °C for 8 weeks.

The RNA was extracted from the whole plant of *A. purpurea* with using FastPure Plant Total RNA Isolation Kit (Polysaccharides & Polyphenolics–rich) (Vazyme, Nanjing, China) according to the manufacturer's protocol. The cDNA was prepared with 5 μg total RNA with a HiScript III 1st Strand cDNA Synthesis Kit (+gDNA wiper) (Vazyme, Nanjing, China).

### 2.2. Cloning of ApNST1, ApNST1.1, ApNST1.2 and ApNST1.3 and Plasmid Construction

*Cynara scolymus* L. is a plant in the Compositae family and is closely related to *A. purpurea*. Based on the *CsNST1* (accession number LEKV01004794, NCBI) CDS sequence downloaded from NCBI, the primers NST1-F (5′-ATGCTGCCCTCTCCTTTGAAT-3′) and NST1-R (5′-GCGAATTTGACCGGATTGG-3′) were designed to amplify *ApNST1*, using cDNA from *A. purpurea* as a template. The PCR fragments were obtained and cloned into plant expression vector *Super35S::GFP* by double digestion technique.

### 2.3. Tobacco Transformation

Plant expression plasmids were transferred into competent cells of the *Agrobacterium tumefaciens* strain GV3101 through freezethaw treatment. The transformed *A. tumefaciens* colonies were selected on LB-agar plates containing 50 mg·L$^{-1}$ of kanamycin, 50 mg·L$^{-1}$ of rifampicin and 50 mg·L$^{-1}$ of gentamicin. The positive colonies were identified using PCR amplification of the inserted genes and were used for the tobacco transformation as previously described [30]. The transgenic plants were confirmed by qRT-PCR (Figure S1). In the follow-up experiment, *Super35S::GFP* was used as the control group.

### 2.4. Quantitative Real-Time PCR Assay

Quantitative real-time PCR (qRT-PCR) was performed using ChamQ Universal SYBR qPCR Master Mix (Vazyme, Nanjing, China) according to the manufacturer's instructions under following conditions: initial denaturation at 95 °C for 30 s, 40 cycles at 95 °C for 10 s and 60 °C for 30 s, At the end of the qRT-PCR cycles, the products were subjected to melt curve analysis to verify the specificity of PCR amplification. Three independent experiments were performed. Relative expression levels were calculated using the $2^{-\Delta\Delta Ct}$ formula with Actin 7 as a housekeeping gene [31]. The used primers are shown in Supplemental Table S1. The expression of transgenic tobacco was significantly higher than that of WT, which demonstrated the successful transfer and expression of four different *NST1* genes in tobacco.

### 2.5. Transgenic Tobacco Growing Conditions

In this study, transgenic tobacco was investigated in a sterile environment and artificial climate chamber environment. First, the seeds were soaked in sterile water, and then the surface of the seeds was sterilized with 75% alcohol for 10 s, washed with sterile water 3 times, sterilized with 2% sodium hypochlorite for 10 min and washed with sterile water 3 times. Then, they were seeded into resistance screening medium with re-suspension, and then placed in an incubator under a light intensity of 35 μmol·m$^{-2}$·s$^{-1}$, 28 °C light for 16 h and 25 °C darkness for 8 h to obtain aseptic seedlings. In this study, Murashige and Skoog medium (MS) containing 50 mg/L hygromycin was used as the resistance screening medium and MS resuspension contained 0.15% agar [31]. MS + 30 g·L$^{-1}$ sucrose + 6 g·L$^{-1}$ agar is a basic medium.

### 2.6. Abiotic Stress Treatment

2.6.1. Salt, Abscisic Acid and Low-Temperature Treatment under Aseptic Conditions

The healthy seedlings were transferred to the corresponding medium for treatment. An MS medium containing 200 mmol·L$^{-1}$ NaCl (a medium stress) and 200 μmol·L$^{-1}$ abscisic acid (ABA) was placed in the light incubator at a light intensity of 35 μmol·m$^{-2}$·s$^{-1}$, 28 °C for 16 h and 25 °C for 8 h for salt treatment and abscisic acid treatment, respectively. At the same time, MS medium was placed in a 4 °C incubator for low-temperature treatment.

2.6.2. Salt, Drought and Low-Temperature Treatment under Natural Conditions

After 15 days of seeding and robust growth, the sterile seedlings were planted into turf soil and treated in the environment of an artificial climate chamber. The soil was watered slightly to moisten the soil, and then shaded for two days to maintain water and take root. When the seedlings grew steadily and robustly, they were treated with salt, drought and low temperature, and the salt concentration was 200 mmol/L. Low-temperature treatment was maintained using a 4 °C low-temperature incubator.

### 2.7. Fresh Weight, Root Length and Stem Height Were Measured

The treated tobacco was weighed to determine the fresh weight of the plant. The root length of tobacco before and after treatment was measured using Image J 1.4.3.67. The same software was used to measure the height of adults before and after treatment and after rehydration.

### 2.8. Sequence Alignment

The amino acid sequences of *ApNST1*, *ApNST1.1*, *ApNST1.2* and *ApNST1.3* were compared. Sequences were compared using DNAMAN (version 8.0, Lynnon Biosoft, Quebec, QC, Canada), MEGA (version 7.0, Mega Limited, Auckland, New Zealand) and Genedoc was used to enhance the figure.

### 2.9. Paraffin Section

The treated transgenic tobacco was taken as the material, and stalks with lengths of 1 cm at 1/3 above the soil surface were taken as samples for treatment. The stalks were placed in FAA (70% alcohol: formalin: acetic acid 18:1:1), fixed for 24 h and then soaked in hydrogen peroxide and glacial acetic acid (1:1). The material was softened in the mixed solution for 48 h, after which, the sample was dehydrated with ethanol and embedded in paraffin for paraffin penetration embedding. The sample was divided into 10 μm sections using a Microslicer (Leica RM-2145). Finally, the sample was stained with saffron-solid green, examined using a microscope and analyzed via image collection. Image J 1.4.3.67 was used to measure xylem thickness and cell wall thickness for statistical analysis.

### 2.10. Analysis of Data

All treatments mentioned in this study involved at least three independent biological and technical replicates. Microsoft Excel 2016, GraphPad Prism 8.0.2 and IBM SPSS 26 were used for data statistical testing and analysis.

## 3. Results

### 3.1. Bioinformatics Analysis of the Four Splice Variants from A. purpurea NST1

We obtained sequence amplification primers from *CsNST1* to clone the *ApNST1* gene. Then, our research group identified four splice variants, which were named *ApNST1*, *ApNST1.1*, *ApNST1.2* and *ApNST1.3*. We compared the amino acid sequences resulting from these four splice variants, and the results are shown in Figure 1a. It was observed that the *ApNST1.1* transcript had splicing sites at the 5th end of one exon, and the *ApNST1.2* transcript had splicing sites at the 3rd end of another exon. *ApNST1* has both of these splicing sites; i.e., *ApNST1* has two splicing types: alternative 3'splicing and alternative 5'splicing, and *ApNST1.1* has alternative 5'splicing. *ApNST1.2* underwent alternative

3′splicing, as shown in Figure 1b. *ApNST1*, *ApNST1.1*, *ApNST1.2* and *ApNST1.3* were compared with the homologous genes in the NCBI database, and it was found that their nucleotide sequences had higher homology with *CsNST1*, the similarity is 58.33–79.21%. Although they did not contain the *NST1* domain, they maintained homology with *NST1*, so the naming method was still adopted in this study. It is worth noting that splice variants *ApNST1* and *ApNST1.2* have premature termination codons (PTCs), and this results in the loss of protein expression (Figure 1a).

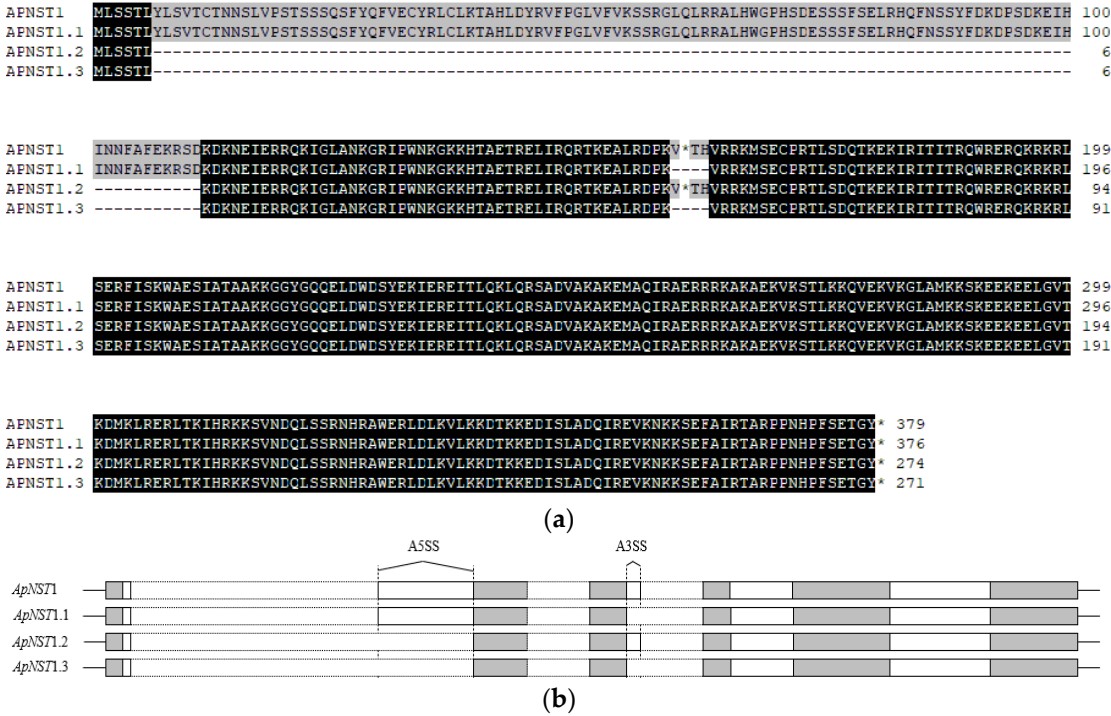

(a)

(b)

**Figure 1.** Biological analysis of four splice variants. (**a**) Amino acid sequence alignment; (**b**) Diagram of alternative splicing mode. Note: The gray part represents the exon and the white part represents the intron.

### 3.2. Phenotypes of Transgenic Tobacco Seedlings under Abiotic Stress

The transgenic tobacco seeds obtained by our research group were sown, and tobacco seedlings with two true leaves and similar growth potential were subjected to abiotic stress, which included treatment with salt, low temperature (4 °C) and abscisic acid (ABA). According to the analysis of phenotype, including the data on weight and root length (Figure 2), under salt stress, *ApNST1*, *ApNST1.1* and *ApNST1.3* all showed a certain salt tolerance. Among these, *ApNST1.3* showed a significant change in fresh weight, compared to itself and to *Super35S::GFP*. However, *ApNST1.2* grew slowly; its fresh weight did not change.

ABA stress inhibited the growth of the plants; the seedlings did not change and the leaves displayed a yellow phenomenon. Overall, ABA also inhibited root elongation, but transgenic plants significantly moderated this effect. We know that ABA is a strong growth inhibitor, as it inhibits cell division and elongation, and can inhibit the growth of whole plants or isolated organs. Under low-temperature stress, the fresh weight of seedlings changed little and showed no cold resistance. Under normal growth conditions, *ApNST1.1* plants grew vigorously, grew quickly and had large leaves.

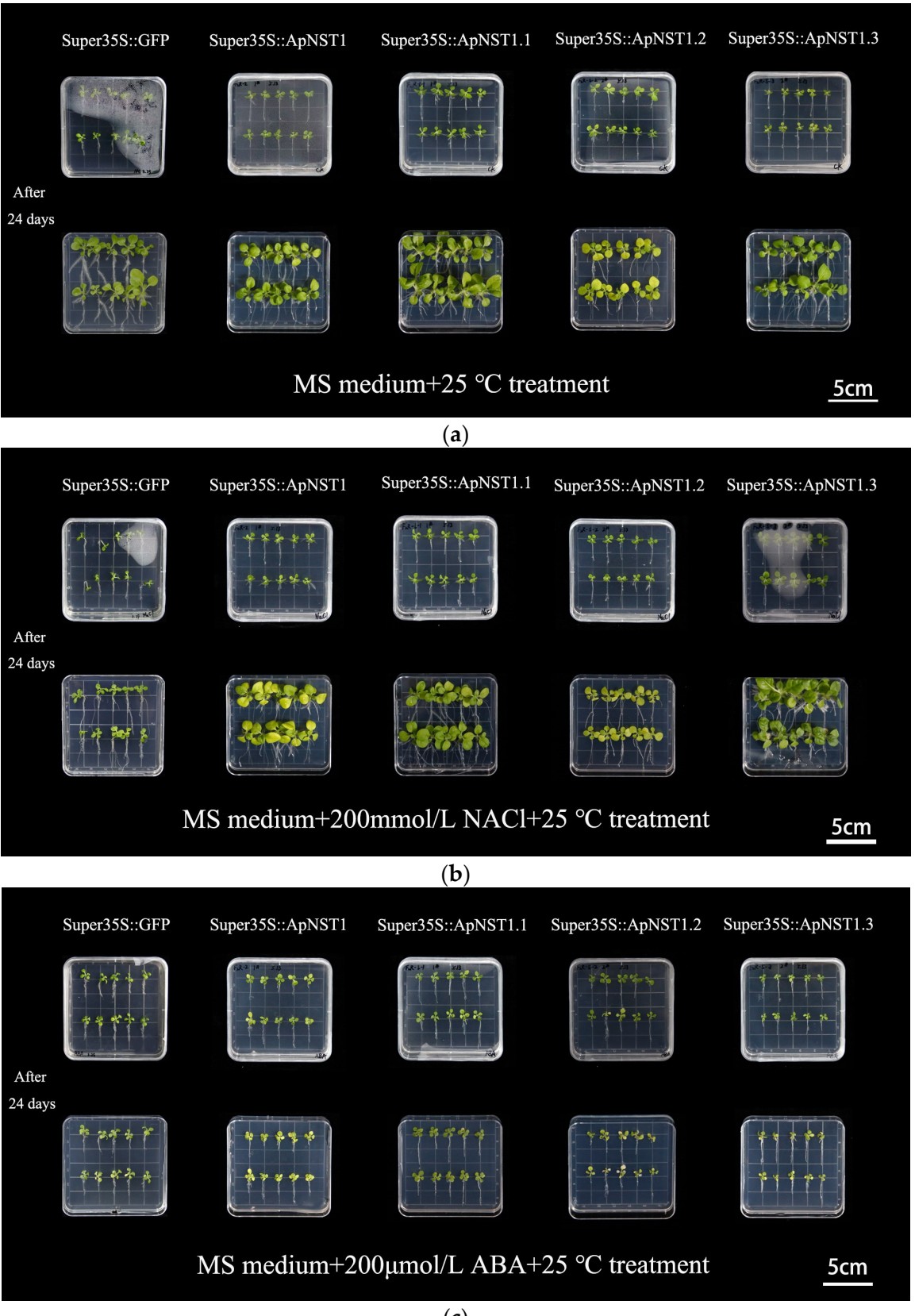

**Figure 2.** *Cont.*

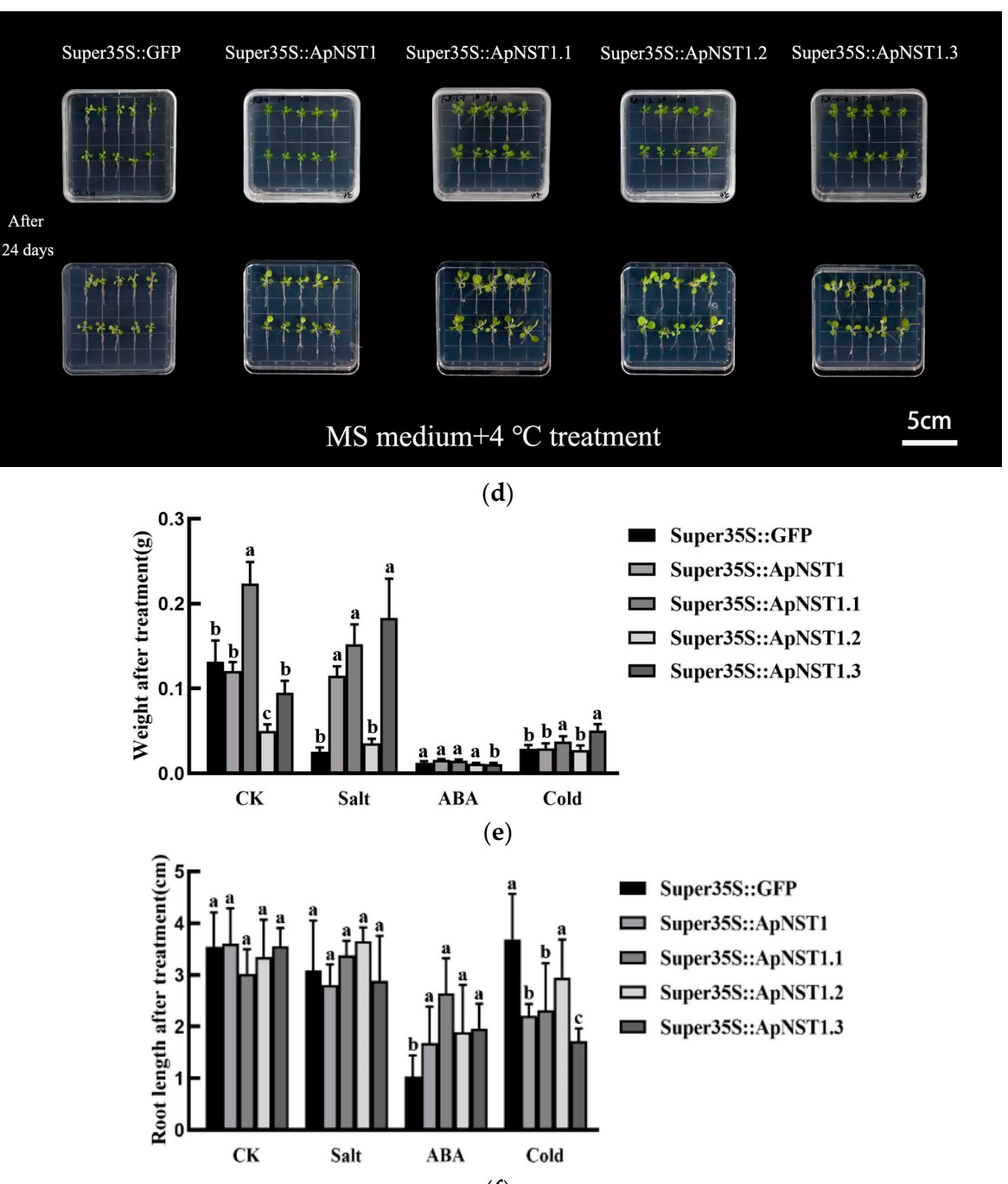

**Figure 2.** Phenotypes and data maps of transgenic tobacco with different splice variants under different abiotic stresses in aseptic environment. (**a**) Phenotypes of transgenic tobacco seedlings in control group; (**b**) Phenotypes of transgenic tobacco seedlings treated with salt; (**c**) Phenotypes of transgenic tobacco seedlings treated with ABA; (**d**) Phenotypes of transgenic tobacco seedlings treated with low temperature; (**e**) Weight data graph of the whole transgenic tobacco after stress treatment; (**f**) Root length data graph of transgenic tobacco after stress treatment. CK: Transgenic tobacco that has not been treated with abiotic stress. The data were analyzed by means of the Waller–Duncan test in Statistical Product and Service Solutions (SPSS) statistical software ($p < 0.05$).

### 3.3. Phenotypes of Transgenic Tobacco Mature Seedlings under Abiotic Stress

The transgenic tobacco seedlings were planted in the artificial climate chamber. When their growth was relatively robust, plants with similar growth were selected to be treated

with salt, low temperature and drought for 15 days, and rewatered for 7 days after treatment. The growth conditions of the plants in the two stages were observed (Figure 3a–d). The average elongation of *ApNST1.1* before and after treatment was 10 cm, which was 1.3 cm higher than *Super35S::GFP* (Figure 3e), and the change of stalk height significantly increased. However, compared to the control group, the lengthening of the other three splice variants was slower and the final elongation was lower, indicating that salt–alkali tolerance of *ApNST1*, *ApNST1.2* and *ApNST1.3* was not obvious. Compared to *Super35S::GFP*, the stem of *ApNST1.2* showed no obvious elongation and showed significant salt tolerance, which was consistent with the observations of the seedling phenotype. However, the transgenic tobacco plants began to grow rapidly after rewatering, indicating that *ApNST1.2* was given appropriate growth conditions after being removed from the high-salt environment. *ApNST1.1* could still recover normal growth, but there was no obvious growth trend after rehydration, indicating that high salt conditions did not affect its growth, and it reached flowering state before rehydration; i.e., vegetative growth was transformed into reproductive growth, and the stem no longer extended significantly.

In the process of low-temperature treatment, the stem height of *ApNST1* and *Super35S::GFP* changed by approx. 2 cm, similar to the change in stem height of *ApNST1* under normal growth conditions. Moreover, the stem elongation rate remained unchanged, indicating that the growth of *ApNST1* was not affected by a low-temperature environment and had certain cold resistance. The average elongation of *ApNST1.2* before and after treatment was 6.1 cm, 4.3 cm higher than *Super35S::GFP*, and the cold resistance was obvious. Low temperature inhibited the growth of *ApNST1.1* and *ApNST1.3*. Under low-temperature conditions, transgenic tobacco plants did not grow significantly after rehydration.

Under drought stress, the rate of stem elongation of *ApNST1*, *ApNST1.1*, *ApNST1.2* and *ApNST1.3* slowed, and all of them recovered after rewatering, indicating that they had no drought resistance.

After the treatment, the plants were rehydrated for 7 days, and it was found that after the water supply was restored, the plants could grow normally and did not die during the treatment (Figure 3f).

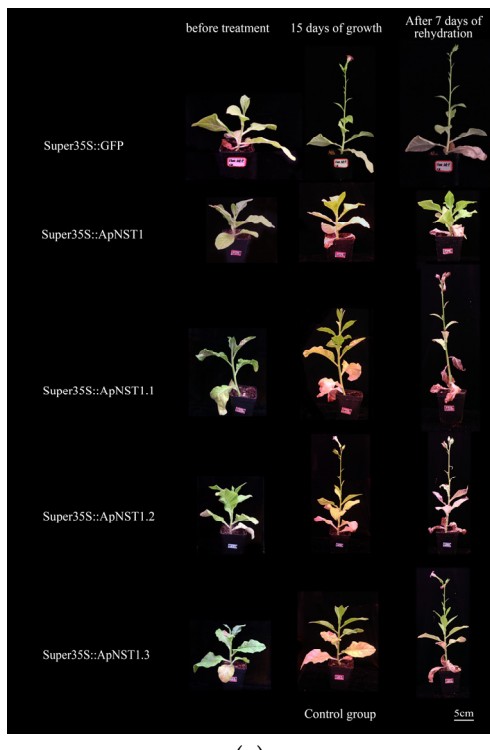

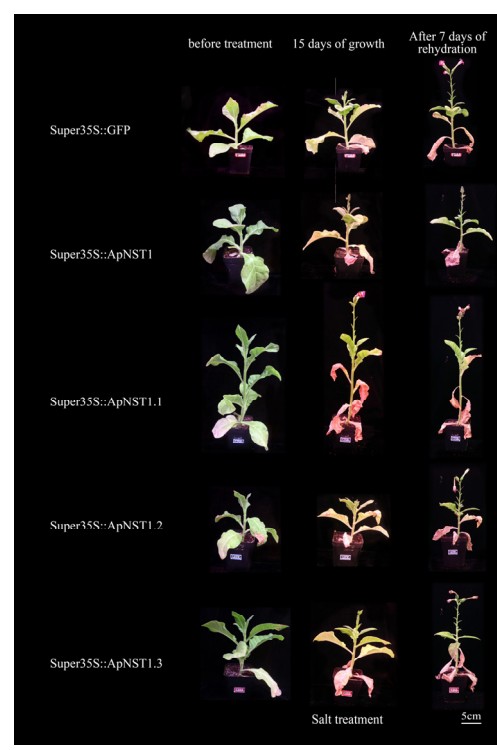

(**a**)                                                                        (**b**)

**Figure 3.** *Cont.*

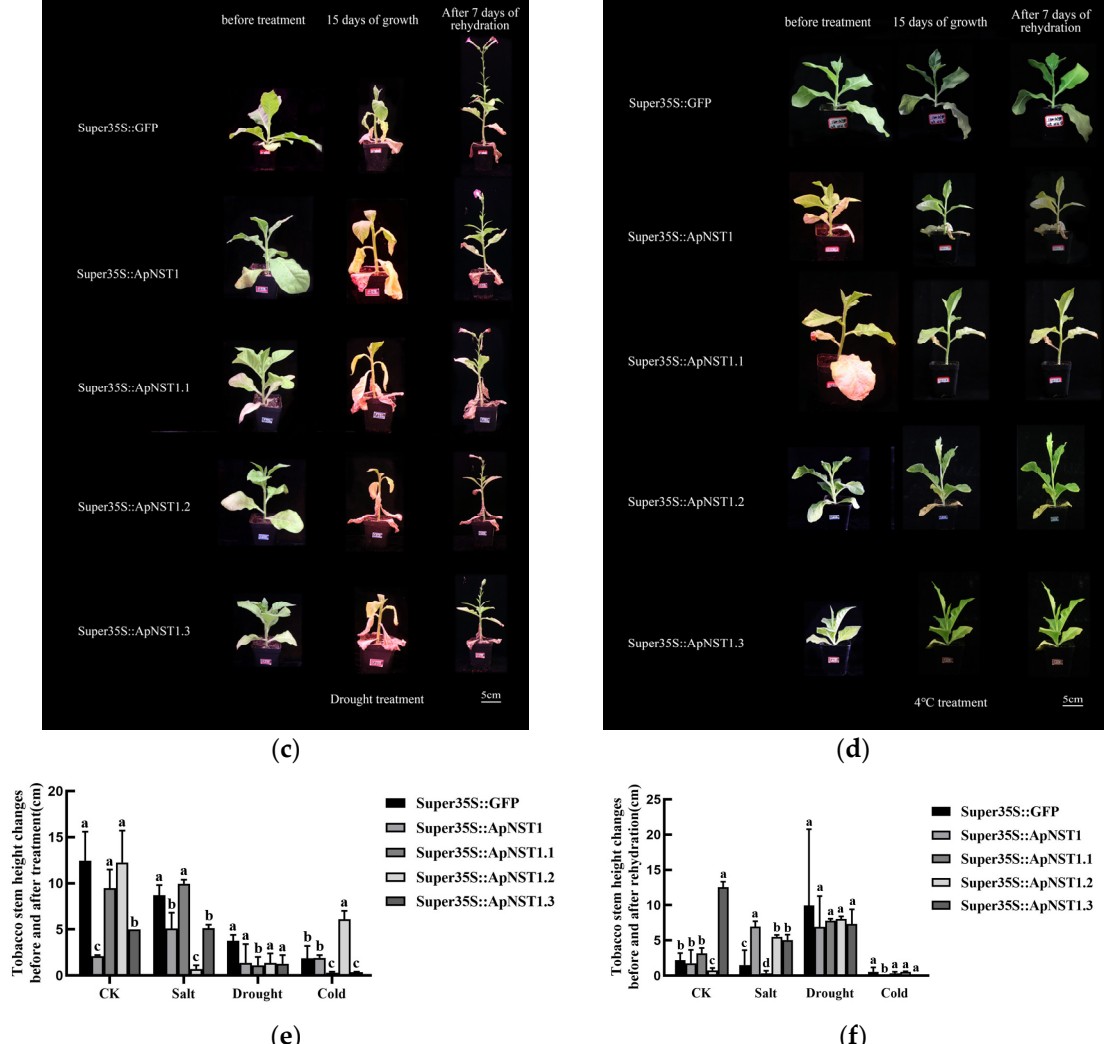

**Figure 3.** Phenotypes and data maps of transgenic tobacco with different splice variants under different abiotic stress in natural environment. (**a**) Phenotypes of transgenic tobacco mature seedlings in control group; (**b**) Phenotypes of transgenic mature tobacco seedlings treated with salt; (**c**) Phenotypes of transgenic tobacco mature seedlings treated with drought; (**d**) Phenotypes of transgenic mature tobacco seedlings treated with low temperature; (**e**) Graph of transgenic mature tobacco stem height variation before and after stress treatment; (**f**) Graph of transgenic mature tobacco stem height variation before and after rehydration. CK: Transgenic tobacco that has not been treated with abiotic stress. The data were analyzed by means of the Waller-Duncan test in SPSS statistical software ($p < 0.05$).

### 3.4. Analysis of Cross-Cut Structure of Transgenic Tobacco Stem

Transection of 12-week-old transgenic tobacco stems showed epidermis, cortex, phloem, xylem and pith. Among them, the parts stained red by casserole solid green were lignified cell walls and tubes, while the parts stained green were plant cellulose cell walls and sieve tubes (Figure 4a–e).

Xylem and cell wall thickness of *Super35S::GFP*, *ApNST1*, *ApNST1.1*, *ApNST1.2* and *ApNST1.3* transgenic tobacco were measured and compared, and statistical analysis is shown in Figure 4f,g. The results show that there were significant differences in xylem and cell wall thickness among the five transgenic tobacco plants. The average xylem thicknesses of *Super35S::GFP* and *ApNST1* were 126 μm and 220 μm, respectively, and the average xylem thickness of *ApNST1.3* is about 30 μm thicker than that of *Super35S::GFP*. Compared to *Super35S::GFP*, the average xylem thickness of *ApNST1.2* was approx. −50 μm, which

was consistent with the salt tolerance of seedlings and adult seedlings. Cell wall thickness changed little, but all of them had thickened, and *ApNST1* was the most obvious in this regard. The thickening of the single cell wall was about 0.215 μm, indicating that *NST1* had played a role. Indeed, it was also evident that multiple types of variable splicing using splice variants increased the impact of the *NST1* transcription factor [32]. The difference in cell wall thickness was very small, which was mainly due to the different number of xylem cells. *ApNST1* had an average of 17 cells in the xylem, while *ApNST1.2* had an average of 5 cells. The number of xylem cells of other splice variants remained at approx. 10. (Figure 4a–e). The xylem thickness of *ApNST1* and *ApNST1.2* exhibits obvious differences that may be related to the fact that they have PTCs. Future studies could investigate the relationship between these factors.

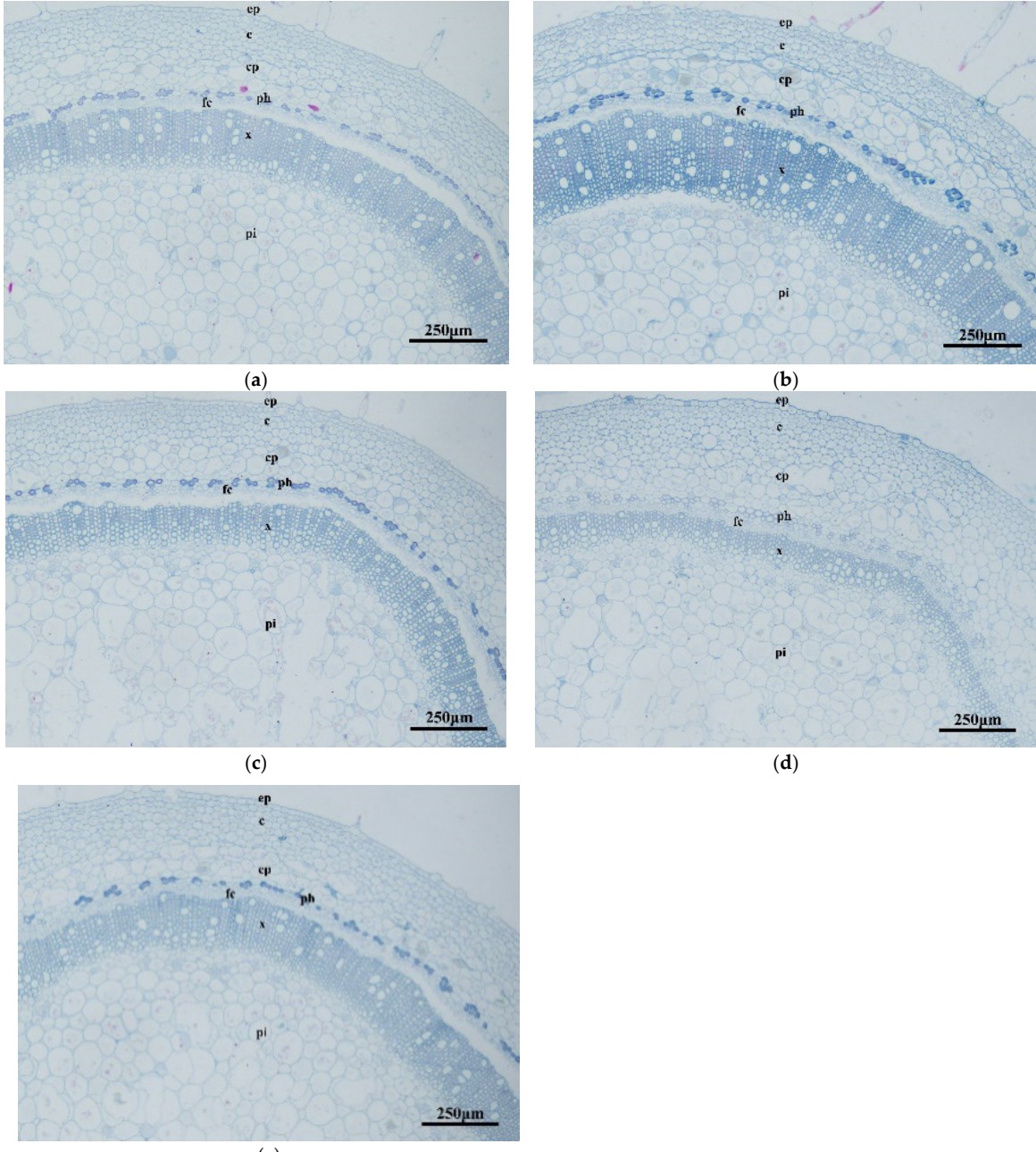

**Figure 4.** *Cont.*

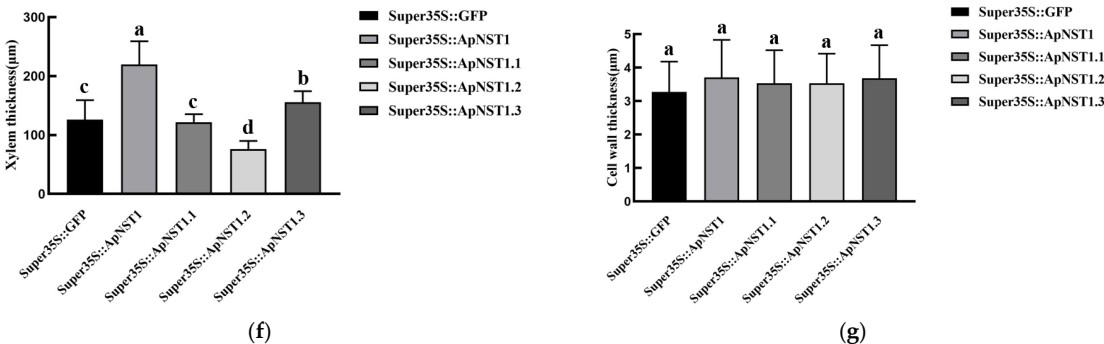

(f)

(g)

**Figure 4.** Cross-section structure and data analysis of transgenic tobacco. (**a**) *Super35S::GFP* stem cross-cutting structure; (**b**) *ApNST1* stem cross-cutting structure; (**c**) *ApNST1.1* stem cross-cutting structure; (**d**) *ApNST1.2* stem cross-cutting structure; (**e**) *ApNST1.3* stem cross-cutting structure. Note: ep: epidermis; c: collenchyma; cp: cortical parenchyma; ph: phloem; fc: cambium; x: xylem; pi: pith. (**f**) Xylem thickness data plot; (**g**) Cell wall thickness data plot between two cells. The data were analyzed by means of the Waller-Duncan test in SPSS statistical software ($p < 0.05$).

## 4. Discussion

AS is an important way to generate notable regulatory and proteomic complexity in eucarya. ES and IR are the most prevalent forms of AS in eukaryotic groups, including plants [33]. The alternative splicing of 5′ and 3′splicing sites is to preserve or splice all or part of the sequence of exons through the selection of 5′splicing donor or 3′splicing recipient [34]. 5′ and 3′alternative splicing sites are also very common in alternative splicing types [35], but there is little progress in the research on these two splicing modes in plants, thus, further attention and research are needed in the future. The splice variants that we obtained were A5SS and A3SS; this research has enriched the chrysanthemum variable splicing types. Alternative splicing can increase the variability and complexity of the transcriptome, and AS has two main results: proteome diversification and regulation of gene expression [36]. AS has been suggested as one of the possible origins of the large phenotypic differences between species, which may be similar to the protein-coding gene pool shared by vertebrates [37]. Thus, alternative splicing is considered a "key step between transcription and translation". The AS process has been identified in many plants; it is a very common and important regulatory component in plant growth and development. We believe that with the continuous improvement of sequencing technology and analytical methods, we will have a deeper understanding of this refined post-transcriptional regulation.

With the development of whole genome sequencing and study on the function of the NAC transcription factor, it has been found that NAC transcription factor is expressed in different developmental stages and tissues of plants, and is closely related to lignin synthesis, growth and development of plants, and the regulatory function of adapting to abiotic and biological stresses [38]. The role of NAC transcription factors in growth and development and stress response has been confirmed in many studies [11,12]. Meanwhile, new NAC family genes and functions continue to be discovered. In 2018, Shandong Agricultural University overexpressed the tomato transcription factor *SlNAC35* gene in tomato, enhancing the cold resistance of a transgenic tomato [39]. Various studies have shown that co-transcription or post-transcription mechanisms are highly induced by abiotic stress and involve a large number of stress-related genes, confirming the importance of alternative splicing in plant performance, adaptability and stress resistance.

In the cloning of the *NST1* gene, our research group found four kinds of splice variants, which were found to be homologous to *NST1* of Thistle of Asteraceae by comparison. They were thusly named *ApNST1*, *ApNST1.1*, *ApNST1.2* and *ApNST1.3* and the function of variable splicing of the *NST1* gene of *A. purpurea* was explored under abiotic stress. Transgenic tobacco seedlings and seedlings were treated to observe their phenotypes through analyses of seedling performance of *ApNST1*, *ApNST1.1*, *ApNST1.2* and *ApNST1.3* under

stresses due to low temperature, drought, ABA and salt. We draw the following conclusions: *ApNST1* seedlings had a certain degree of salt resistance, and the adult seedlings had a certain degree of low-temperature resistance. The adult seedlings and seedlings of *ApNST1.1* showed obvious salt resistance. The adult seedlings of *ApNST1.2* showed obvious low-temperature resistance, and both the seedling and the adult seedlings showed significant salt tolerance, but the root elongation at the seedling stage was large. *ApNST1.3* seedlings showed significant salt tolerance, while adult seedlings showed no salt tolerance. The results of resistance at seedling stage and adult seedling stage were inconsistent. We deduced that the seedling stage was a process of nutrient accumulation, or the nutrient conditions given by the medium under aseptic conditions might not be conducive to its growth and could not respond in a timely fashion in the face of biological stress. Therefore, our conclusions were mainly based on the natural soil conditions, and were, namely, that *ApNST1.1* has salt resistance and *ApNST1.2* is resistant to low temperature.

By observing the cross section of transgenic tobacco, it was found that the xylem of *ApNST1* was significantly thickened, and the cell wall was also thickened but not significantly, mainly due to the increased number of xylem cells, indicating that *NST1* played a role. In combination with the study of Liu et al. [3], we believe that *NST1* is related to the formation of the plant secondary cell wall, and ABA phosphorylation of *NST1* promotes the downstream reaction. The formation of secondary cell wall is conducive to plant growth, and overexpression of *NST1* can achieve this effect. The characteristics of adult seedlings and seedlings were not completely consistent. Excluding the potential for slight errors in manual measurement, we assume that the functional expression of *NST1* gene may be repeated by other genes, which will be explored and studied later [40]. The xylem of *ApNST1.2* is obviously thinner, but it can respond to low-temperature stress. We suspect that xylem thickness varies in response to abiotic stress, especially temperature stress, and this idea remains to be investigated. In addition, the growth cycle of *A. purpurea* was long and its rooting was slow; thus, due to time restrictions, this study was unable to conduct abiotic stress treatments on the transgenic *A. purpurea*. Instead, transgenic tobacco was selected for this study. However, the observation and treatment of transgenic chrysanthemum are also being carried out.

## 5. Conclusions

In this study, we analyzed the amino acid sequence of the four splice variants from *A. purpurea NST1* and found that *ApNST1.1* had A5SS and *ApNST1.2* had A3SS, and *ApNST1* had the two splicing types. We verified the different resistance effects of the four splice variants in transgenic tobacco, and we concluded that *ApNST1.1* has salt resistance and *ApNST1.2* is resistant to low temperature. The cross-cut structure of transgenic tobacco stem was observed by paraffin section technology to determine the changes in xylem thickness and the reasons for the changes. We believe that the xylem thickness is related to the growth rate, and the xylem thickness of stems with a slow growth rate is obviously thicker, while the xylem thickness of stems with a fast growth rate is not, which is consistent with the change of stem height (Figure 3a,e). However, the relationship between the xylem thickness and non-stress treatment is still unknown, which will be something that we need to pay attention to in future research. In this study, the splice variants with obvious salt and cold resistance were selected, which enriched the germplasm resources of chrysanthemum. However, the mechanism of action in *A. purpurea* is still unclear, more time is needed to verify the role of these four splice variants in *A. purpurea*. In the future, we hope to investigate this and identify the most resistant *A. purpurea*. The objective of our research is to verify whether the obtained *NST1* splice variants have a resistance function and to identify resistant materials. This study initially investigated which splice variants plays a role in resistance functions. Our long-term goal is to improve the adaptability of *A. purpurea* to allow its ornamental application in many areas. The functional identification of transgenic tobacco in this paper has laid a foundation for our long-term goal, and provided the possibility for obtaining clear resistance to stressors in *A. purpurea*.

**Supplementary Materials:** The following supporting information can be downloaded at: https://www.mdpi.com/article/10.3390/horticulturae9080916/s1, Figure S1: Relative expression of transgenic tobacco; Table S1: List of the primers used in the analyses of gene expression of *NST1* by qRT-PCR.

**Author Contributions:** Conceptualization, H.W. and X.H.; methodology, H.W. and X.H.; software, X.H.; validation, Y.G., W.Z. and X.H.; formal analysis, W.H. and Y.L.; investigation, X.Z. and S.C.; resources, H.W.; data curation, X.H.; writing—original draft preparation, X.H.; writing—review and editing, H.W. and X.S.; visualization, X.H.; supervision, H.W.; project administration, H.W.; funding acquisition, H.W. and X.S. All authors have read and agreed to the published version of the manuscript.

**Funding:** This research was funded by National Natural Science Foundation Youth Fund Project (32101580), Research Fund for High-level Talents in Qingdao Agricultural University (6631122019), Survey and Collection Project of Herbal Plant Germplasm Resources at Shandong Provincial Forest and Grass Germplasm Resources Center (6602423134).

**Data Availability Statement:** Data is contained within the article or Supplementary Material.

**Conflicts of Interest:** The authors declare no conflict of interest.

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
