# Peer review of "Functional Verification of the Four Splice Variants from Ajania purpurea NST1 in Transgenic Tobacco"

_horticulturae, doi:10.3390/horticulturae9080916_

Round 1

Reviewer 1 Report

Dear Authors,

It was a pleasure to read your manuscript. "Functional verification of the four alternative spliceosomes of 2 NST1 in Ajania purpurea".

I feel the manuscript should undergo some major revision based on the suggestions cloning strategies of NST1 genes and the phenotypic correlation between NaCl stress under aseptic conditions and natural conditions.

The manuscript written well but it has lack of results interpretation and the correlation between NaCl treatment and NST1 gene and also there is a lack of information about the clones of NST1 from their previous study (Line no 158-159). If the previous clones of NST1 are published, they can provide the reference or they can add the cloning and vector construction details in materials and methods section (Line 175-176). All the citations are relevant and support to the study. The manuscript is scientifically sound and contributes to resistance breeding and abiotic stress to crops. The work was designed well and lack of scientific supportive information provided, all the results presented in reproducible, and the details given in the methods sections are appropriate. English writing is understandable.

There are some areas for major revision/suggestions to be considered before consideration for the acceptance.

Query 1:

Line no 14-15

Can be combined as one sentence.

Query 2:

Figure no 2

Salt means Sodium Chloride, it should write as NaCl not NACL in Fig.2 (b). and give space as 200mmol/L NaCl.

Fig 2 (c) 200µmol/L ABA (Space)

Figure 2 can be arranged vertical (One by one). It is difficult to see the plants in present form.

Query 3:

Phenotypic correlation between abiotic stress and natural environment should be discuss more. with supportive evidence with qRT-PCR results.

Line no 331: can be mentioned specifically with what exactly the resistance effects observed.

Query 4:

Line no 332-333

To the Abiotic resistance research work phenotypic variations of root length and plant height is enough to conclude the resistance of NST1 gene. Those variations should correlate with the gene fold or expression assays by qRT-PCR.

Query 5:

Line no 334-335

What is the connection between abiotic treatment and xylem thickness ? The reason should be clearly mentioned in the conclusion section.

Query 6:

Line no 338-339 we need more time to verify the role of these four spliceosomes in Ajania purpurea.

Need clarity of this sentence. Whether the transgenic lines from previous study compared with abiotic stress lines. There is a huge lack of connections between the transgenic lines of NST1 from previous study and the abiotic stress lines from current study.

Author Response

请参阅附件

Reviewer 2 Report

This study is about the function verification of the four spliceosomes of a gene called NST1, which is originated from Ajania purpurea, by using the tobacco plants as the transgenic host.

1) The title should be amended to best suit the study, which used transgenic tobacco as the host for verifying the function of the NST1 gene from Ajania purpurea.

2) Line 77-90, this paragraph is directly copied and pasted from another article (https://doi.org/10.3389/fpls.2022.1120961), we do not encourage partial plagiarism in scientific writing.

3) The methods of how the authors construct the transgenic tobacco plants containing each spliceosome are missing. The authors should also state clearly that Super35S::GFP is a reference control for the experiment.

4) Line 114, bacterium-free vaccine?

5) The accession numbers for the NST1 gene and proteins must be provided.

6) Please double-check all the abbreviations, when they are first shown, they should be written in full . For instance, Line 114 MS, Line 118 ABA, Line 262 AS.

Reviewer 3 Report

The manuscript titled “Functional verification of the four alternative spliceosomes of NST1 in Ajania purpurea” attempts to demonstrate the different functions of 4 proteins encoded for the same gene, but generated from different alternative splicing events. Although the results are interesting, there is a profound lack of details throughout the whole manuscript that makes it hard to evaluate the experimental design and the conclusions. I will focus only on what I consider should be improved in order to be confident that the conclusions drawn in this manuscript are valid.

The title of the manuscript should be corrected because it doesn't reflect what it is about. What the authors are reporting is the heterologous expression of splicing variants from the NTS1 gene from Ajania purpurea in tobacco plants, rather than the analysis of a spliceosome, which is an RNA/protein complex. Similarly, throughout the whole manuscript, the term spliceosome is mainly used incorrectly.

This manuscript is based on the results that the authors obtained from the cloning of the NTS1 gene from Ajania purpurea (line 94), but there is not either a detailed explanation about how the splicing variants were cloned (from which tissue, under which conditions, age of the plant) or a reference to another manuscript that can lead the reader to that information. Furthermore, there are no details of how the transgenic tobacco plants were generated (which protocol, which vector, validation of the positive clones, etc), or how many lines were analyzed during this experiment.

Authors need to demonstrate that the tobacco plants are indeed expressing the transgene, since the differences in the response may be due to differences in expression rather than differences in the protein function. 

None

Reviewer 4 Report

Revision required

Reviewer 5 Report

The study on Ajania purpurea's NST1 gene splice variants and their response to abiotic stress presents intriguing findings. The plant's ability to actively regulate stress response genes and transcripts to survive adversity is remarkable. The identification of four variable splice variants of NST1 gene in Ajania purpurea sheds light on the importance of alternative splicing in eukaryotic gene transcription and its role in various biological processes. The research demonstrates that ApNST1.1 and ApNST1.3 exhibit significant salt tolerance in seedlings, while ApNST1.2 shows notable cold resistance. These findings hold great promise for breeding and applying ApNST1 splice variants with enhanced stress resistance in horticulture and contribute to the study of abiotic stress resilience. Nonetheless, there are some issues that need to be addressed before publishing the paper:

-          Please, provide the initials of the author of the species’ name when mentioning it for the first time

-          Keywords should be arranged alphabetically. Moreover, please, do not repeat words from the title.

-          Why are you mentioning chrysanthemum in the Abstract and Introduction? I know that these two species are related but it may not be obvious for other readers.

-          In the Introduction, refer to plants, not insects or humans.

-          It is not true that “plants cannot move”.

-          In the Materials and methods, you never explain the light conditions (intensity, color and source).

-          References are missing, e.g. for the MS medium or qRT-PCR procedure.

-          More details on the producers of key chemicals, equipment and software are missing (i.e. city, state, and country)

-          Unit style is incorrect. It should be mg·L-1 (not mg/l).

-          In the Results section do not repeat information form the Materials and Methods.

-          Lines 214-217 should be transferred to the Discussion.

-          Figure 2 (legend): weight of what?

-          Figure 3: what treatment?

-          “The role of NAC transcription factors in growth and development and stress response has been confirmed by more and more studies”. References are missing.

-          The first sentence of the Conclusions is unclear and needs to be rewritten.

-          MDPI uses serial comma.

-          References should be formatted in accordance with the MDPI style.

The language is generally understandable, although minor grammar, punctuation, and style corrections are needed.

Reviewer 6 Report

In this study, four alternative splice variants of NST1 gene, and their molecular mechanism in abiotic stress were identified from Ajania purpurea. ApNST1, Ap- 18 NST1.1 and ApNST1.3 showed salt tolerance at seedling stage, and ApNST1.1 showed obvious salt tolerance, while ApNST1.2 showed cold resistance at mature seedling stage. Did You investigate the rate of lignification in the transgenic plants? Can this have negative side-effects?

The graphs are sufficiently detailed and easy to expound. The text contains occasional typos, but it is nevertheless easy to read. Please use proper scientific expressions.  The English of the manuscript needs to be excessively improved. The literature used is enough. All Latin names should be in Italics. ‘Et al.’ is usually in Italics. Check the citations in the running text (insert space between the numbers). Insert space between numbers and the unit. Capitalize all selection gents, or none. Avoid naming researchers in the text. Double check all capitalizations and spaces in the text. Figure 1/a and 1/b takes too much space, consider to remove. A simple similarity index would be enough (at nucleic acid and amino-acid level). At Figure 2/a-d, please indicate, which plantlets are the fresh seedlings, and which ones are the mature seedlings.  The relative expression of which gene is on Figure 2/d?  Please add the scientific name of the tobacco, that You used. 

The English of the manuscript needs to be excessively improved.

Reviewer 7 Report

The manuscript deals with the following on from previous work on the NAC gene family. The authors attempted to verify the splice variants already observed for their functional role.

The manuscript is well written and illustrated, the used methodology and methods are current.

TWo points need to be improved

1- statistical analyses are not clearly presented. The means comparison test used is not presented. Is it Fisher, Duncan, Newmann & Kuels?

2- it is admitted that under controlled conditions (in vitro methods) that a stress of 200 mmol/L NaCl is considered as a medium stress. therefore, it can be suggested that the conclusions must be drawn with caution.

Round 2

Reviewer 1 Report

Dear Authors,

 It was a pleasure to read the revised manuscript entitled "Functional verification of the four alternative spliceosomes of NST1 in Ajania purpurea".

 I hope the authors are clearly gone through all the comments and changes made to correct it on the revised manuscript. I am satisfied with the author's responses. Thanks to all authors for considering my suggestions on this manuscript.

 Thank you

Author Response

Dear reviewer,

Thank you for your comments concerning our article entitled- Functional verification of the four alternative spliceosomes of NST1 in Ajania purpurea(Manuscript ID: horticulturae-2453204). Revised portion are marked in yellow in revised paper. The main corrections in the article and the responds to the reviewer’s comments are as flowing:

Comment:

 I hope the authors are clearly gone through all the comments and changes made to correct it on the revised manuscript. I am satisfied with the author's responses. Thanks to all authors for considering my suggestions on this manuscript.

Response: Thank you for your careful reading of our article. We have reconsidered your suggestion and revised part of the content of the manuscript, adding the description of Figure 2f in the section of Materials and methods. (Line 122-132). 

We tried our best to improve the article and made some changes in the article. As for the problems you mentioned, we have been revised and marked in yellow in the manuscript. We appreciate for Reviewers’ warm work earnestly, and hope that the correction will meet with approval.

Once again, thank you very much for your comments and suggestions.

Reviewer 3 Report

The authors have modified the manuscript to address my previous comments. However, there are a few areas of opportunity to increase the readability of their work. I have one major comment and some minor comments that will allow the reader a better understanding of this work.

Major comment:

The authors have added a plot showing the relative expression of 4 different NTS1 splice variants, figure 2g, however, there is no description either in the results or in the methods section of how the analysis was carried out. Because of that, there are several questions with no answer, for instance, since the WT plant does not have the NTS1 transgene from A. purpurea, what do the authors are quantifying in the WT sample and in the transgenic plants?  Finally, figure 3e and 3f have the same name and same description at the bottom of the plot, but different results. Why are those 2 different? And why is one of them not explained in the manuscript?

The title might need the word variants “... four splice variants from …”

Line 8: It's likely that the sentence was not finished, “Its corolla is purplish red 8 from the middle to the middle” since line 31 reads “ Its corolla is purple-red from the middle upward”.

Line 11: It's likely this sentence was not finished, “Alternative splicing is a regulatory process” since the idea is not followed up in the next sentence. 

Line 91: “four splice variants of tobacco” change “of” for in.

Line 114 and 116: A. tumefaciens needs to be italicized

Line 133: inoculants, perhaps it refers to seedlings.

Line 144: The word “culture” is not necessary. 

Line 148: At this time, probably the plants are not seedlings anymore, I consider it would be better to call them adults.   

Line 170: The spliceosome term is not correctly used, and the species name is not Anthracia chinensis

Line 171: It can be improved. For instance, instead of using the common name (Thistle), it can be replaced with the scientific name. It should be clarified that the Thistle sequence was used to design primers to clone the gene (such as the authors did in the methods sections) since the current sentence is confusing.

Line 180-181 Figure 1a is about nucleotide alignment but in the text it refers to amino acid alignment. On the other hand, When authors mention “Although it did not contain the NST1 domain”, does it refer to any particular splice variant? If so, which one?

Line 186: Amino acid sequence alignment instead of blast.

According to Figure 1, splice variants 1 and 1.2 have premature stop codons. I think this would be interesting to mention in the main text.

Line 222-224: Please correct this sentence, it is hard to understand. “ ApNST1, ApNST1.2, and ApNST1.3 all weakened”, since it may be interpreted differently as they grew shorter than others. 

In Figure 3, please describe what the letters are (pi, ph,x, etc).

Line 285: I consider this line is not relevant for the discussion of this manuscript, which is about plants “AS is an important way to generate notable regulatory and proteomic complexity in metazoans”

Line 329: “The results of resistance at seedling stage and seedling stage were inconsistent”

In line 362, the word “spliceosome” is not correctly used here, and the following appearances.

Line 364: “However, the mechanism of action of Ajania purpurea”, in Ajania purpurea.

There are several sentences that are not finished or need a connector between ideas.

Reviewer 5 Report

The authors have revised the text and addressed most of my queries, although there are still two issues that need to be clarified:

1) When I wrote "Please, provide the initials of the author of the species’ name when mentioning it for the first time", I meant Ajania purpurea C.Shih. This suggestion refers to all species mentioned for the first time, e.g. Arabidopsis thaliana (L.) Heynh.

2) Lux is an outdated, non-informative unit. Please provide the PPFD value.

English is fine

Author Response

Dear reviewer,

Thank you for your comments concerning our article entitled- Functional verification of the four splice variants from Ajania purpurea NST1 in transgenic tobacco(Manuscript ID: horticulturae-2453204). Those comments are all valuable and very helpful for revising and improving our paper. We have studied comments carefully and have made correction which we hope meet with approval. Revised portion are marked in yellow in revised paper. The main corrections in the article and the responds to the reviewer’s comments are as flowing:

Comment 1: When I wrote "Please, provide the initials of the author of the species’ name when mentioning it for the first time", I meant Ajania purpurea C.Shih. This suggestion refers to all species mentioned for the first time, e.g. Arabidopsis thaliana (L.) Heynh.

Response: Thank you for your advice. I'm sorry I didn't get what you meant when you first mentioned the " Please, provide the initials of the author of the species’ name when mentioning it for the first time ". Thanks for your reminding, now it has been added in the article. (Line 30、40)

Comment 2: Lux is an outdated, non-informative unit. Please provide the PPFD value.

Response: Thank you for your valuable advice. We have changed it to “35 μmol·m-2·s-1” (Line 141、150)

We appreciate for Reviewers’ warm work earnestly, and hope that the correction will meet with approval. Once again, thank you very much for your comments and suggestions.

Reviewer 6 Report

The paper imporved a lot, therefore I support to accept it in present form. 

Author Response

Dear reviewer,

Thank you for your comments concerning our article entitled- Functional verification of the four splice variants from Ajania purpurea NST1 in transgenic tobacco(Manuscript ID: horticulturae-2453204). Thank you for your recognition.